# Structural Behavior of Amphiphilic Triblock Copolymer P104/Water System

**DOI:** 10.3390/polym15112551

**Published:** 2023-05-31

**Authors:** Edgar Benjamín Figueroa-Ochoa, Lourdes Mónica Bravo-Anaya, Ricardo Vaca-López, Gabriel Landázuri-Gómez, Luis Carlos Rosales-Rivera, Tania Diaz-Vidal, Francisco Carvajal, Emma Rebeca Macías-Balleza, Yahya Rharbi, J. Félix Armando Soltero-Martínez

**Affiliations:** 1Departamento de Química, Universidad de Guadalajara, Blvd. M. García Barragán #1451, Guadalajara 44430, Jalisco, Mexico; benjamin.figueroa@academicos.udg.mx (E.B.F.-O.); ricardo.vaca@alumnos.udg.mx (R.V.-L.); 2Université Grenoble Alpes, CNRS, Grenoble INP (Institut of Engineering Univ. Grenoble Alpes), 38000 Grenoble, France; gabriel.landazuri@academicos.udg.mx (G.L.-G.); yahya.rharbi@univ-grenoble-alpes.fr (Y.R.); 3Departamento de Ingeniería Química, Universidad de Guadalajara, Blvd. M. García Barragán #1451, Guadalajara 44430, Jalisco, Mexico; carlos.rosales@academicos.udg.mx (L.C.R.-R.); taniadzv@hotmail.com (T.D.-V.); emmarebecamacias@hotmail.com (E.R.M.-B.); 4Université de Rennes, Institut des Sciences Chimiques de Rennes, Équipe CORINT, CNRS, UMR 6226, Campus de Beaulieu, Bat 10A, 35042 Rennes Cedex, France; 5Centro Universitario UTEG, Departamento de Investigación, Héroes Ferrocarrileros #1325, Guadalajara 44460, Jalisco, Mexico; 6CUTonalá, Departamento de Ingenierías, Universidad de Guadalajara, Nuevo Periférico # 555, Ejido San José Tatepozco 45425, Jalisco, Mexico

**Keywords:** P104, spherical and elongated micelles, rheological behavior, phase diagram

## Abstract

A detailed study of the different structural transitions of the triblock copolymer PEO_27_–PPO_61_–PEO_27_ (P104) in water, in the dilute and semi-dilute regions, is addressed here as a function of temperature and P104 concentration (C_P104_) by mean of complimentary methods: viscosimetry, densimetry, dynamic light scattering, turbidimetry, polarized microscopy, and rheometry. The hydration profile was calculated through density and sound velocity measurements. It was possible to identify the regions where monomers exist, spherical micelle formation, elongated cylindrical micelles formation, clouding points, and liquid crystalline behavior. We report a partial phase diagram including information for P104 concentrations from 1 × 10^−4^ to 90 wt.% and temperatures from 20 to 75 °C that will be helpful for further interaction studies with hydrophobic molecules or active principles for drug delivery.

## 1. Introduction

Amphiphilic block copolymers are synthetized to self-assemble in aqueous solvents as different structures such as spheres and cylinders, among others [1,2,3,4]. AB diblock and ABA triblock copolymers, which consist of only two components, are the two most studied and characterized molecules in the linear block copolymers [5]. However, more recently, some studies have been also emphasized ABC triblock copolymers for their specific bulk morphologies, which present several differences from those observed in linear diblock copolymers through their higher complexity [6]. Amphiphilic triblock copolymers, also known by their commercial name Pluronic^®^, consisting of poly(ethylene oxide) (PEO) and poly(propylene oxide) (PPO) blocks as follows: PEO_x_–PPO_y_–PEO_x_, have been studied due to their efficiency during drug delivery processes [7,8,9,10]. Drug incorporation into the micelle core formed by these amphiphilic block copolymers can provide metabolic stability, superior drug circulation time, and an increment of solubility [9,11]. Therefore, micelle core-shell structure and characteristics are essential for their efficiency in drug delivery [12]. This core is also a compartment incompatible with water but is able to store different kinds of therapeutic reagents [11,13]. Gene delivery is another pharmaceutical application that has been proposed for several block copolymers [9,14,15,16,17]. Furthermore, interactions between multidrug-resistant cancer cells and triblock copolymer unimers have also been studied, resulting in the sensitization of the cells to diverse anticancer agents [18,19].

All these applications depend on the amphiphilic block copolymer structure. In water, they are able to form micelles above the critical micellar temperature (CMT) and the critical micellar concentration (CMC) [2,10]. In these amphiphilic triblock copolymers, it has been shown that the CMC decreases promptly as temperature increases, as well as the CMT diminishing as the copolymer concentration increases [1,2,4]. Furthermore, it has been shown that spherical micelles grow to form worm-like micelles with increasing temperature and amphiphilic triblock copolymer concentration. A drastic viscosity increase of several orders of magnitude is an important characteristic of this growth process [20,21]. The phase behavior of these triblock copolymers in water is particularly dependent on temperature and the relative block size [22]. Furthermore, at higher concentrations and temperatures, these copolymers tend to form a numerous variety of lyotropic liquid crystals [23,24].

The selection of a Pluronic as the best delivery system seems to be dependent on the drug involved [18]. There is now a great choice in the literature, depending on the drug solubility and the release properties of Pluronics [19]. Nevertheless, only a few studies have examined micellar structure changes of Pluronic micelles on the uptake of drug molecules [7,8,9,10,19].

Particularly, P104 amphiphilic triblock copolymer has been one of the most investigated copolymers among the Pluronic family due to its rich versatility [25,26,27,28,29,30,31]. It is considered as a potential candidate for drug delivery [10,32,33,34], for solubilization and encapsulation [35,36,37,38,39], emulsification [40,41,42,43,44], as well as a template for the elaboration of organosilicon mesostructures [28,29,45,46].

All these applications require fine control of the different copolymer structures and the transitions between them. Thus, several studies have dealt with many aspects of the P104 properties, including its self-assembly [47,48,49,50,51,52], its overall structure [53,54,55] as well as the core and the corona morphologies [37,54,56,57,58]. The interactions between the core and the corona and the surrounding aqueous phase (hydration) was also investigated particularly in the diluted case [59]. Some aspects of the rheology of P104 copolymer were also investigated in the concentrated regime. These investigations were carried out using numerous methods including light scattering [37,55,60], neutron scattering [61,62], IFTR [63,64,65], fluorescence [55,66,67,68], and calorimetry [55,69]. Yet, since P104 is a commercial copolymer, its properties are affected by impurities as well as PPO and PEO polydispersities. Therefore, these studies should be performed on the same batch of polymer in the same state.

Small-angle neutron scattering (SANS) measurements have been used to study structural changes in a series of Pluronics, including P104, a moderately hydrophilic ethylene oxide–propylene oxide triblock, upon addition of increasing amounts of the drug ibuprofen, showing that the addition of ibuprofen to P104 reduces the CMT from approximately 20 °C to below 13 °C [19]. Scattering and rheology studies on micellar characteristics of P104 in the presence of anthranilic acid (AA) revealed important variations by changing the pH solution [70]. Furthermore, P104 has also been used in the study of dynamics between Pluronic micelles and liposomes for the vectorization of hydrophobic molecules using a fluorescence technique [11]. The exchange dynamics between both of them, studied at different liposome concentrations, were demonstrated to be a collective mechanism characterized by presenting two rate constants: one rate independent of liposome concentration and the second one with dynamics linearly dependent on liposomes concentration [11].

The aim of this work is to present a detailed study of the formation of micellar structures and morphology of PEO_27_–PPO_61_–PEO_27_ Pluronic (P104) in aqueous solution. This study was performed in the dilute and semi-dilute regions using rheology, dynamic light scattering (DLS), polarized light microscopy, densimetry (*ρ*), sound velocity (*U_s_*), viscosimetry, rheometry, and turbidimetry measurements. With the obtained results, it was possible to determine the transition zones between monomers and micelles, i.e., where P104 monomers are in thermodynamic equilibrium with P104 micelles, followed by the formation of spherical micelles (CMT), the sphere-to-rod micellar transition (GMT), the cloud point temperature (CPT), soft gel and hard gel transitions, as well as the liquid crystalline behavior. Here we report a phase diagram including information for P104 concentrations from 1 × 10^−4^ to 90 wt.% and temperatures from 20 to 75 °C that will be helpful for further studies, applications and interaction studies with hydrophobic molecules, or active principles for drug delivery.

## 2. Materials and Methods

### 2.1. Triblock Copolymer P104/H_2_O Solutions Preparation

Triblock copolymer, Pluronic^®^ P104 [(PEO)_27_–(PPO)_61_–(PEO)_27_] was obtained from BASF and was used as received. This Pluronic^®^ has a Mw = 5900 kg/mol. Water was drawn from a Millipore Milli-Q purification system.

Samples were prepared by weighing suitable amounts of P104 and water in 50 mL glass vials within a concentration range from 1 × 10^−4^ to 90 wt.%. Each sample was left at the temperature of measurement during 24 h to reach the equilibrium. The glass vials were covered with aluminum foil to prevent light degradation of P104 solutions.

### 2.2. Crossed-Light Polarized Microscopy

P104 samples with concentrations from 20 to 95 wt.% were observed with an Olympus BX51 light polarizing microscope (Olympus, Tokyo, Japan) with an objective of 4× and a Q Imagine Camera (Meyer Instrument, Houston, TX, USA) or a Leica DMLM polarizing microscope (Leica Microsystems, Wetzlar, Germany) with an objective 4× and a 3.0 MP Moticam (Kowloon, Hong Kong), between crossed polarizers at different temperatures. The samples were placed on a glass slide and covered with a cover slip, previously cleaned with ethanol, and carefully washed in distilled water.

### 2.3. Viscosity Measurements

An automatic viscometer AMVn from Anton Paar (Graz, Austria) was selected to carry out viscosity measurements for P104 concentrations from 1 to 20 wt.% at temperatures from 10 up to 50 °C. Measurements were performed from the highest temperature to the lowest temperature. The temperature sweeps were performed by taking viscosity values every two degrees in the temperature range from 50 to 10 °C. The sample was stabilized during 5 min at each temperature change. All samples were analyzed at different inclination angles, i.e., 30°, 50° and 70°; and 5 replicates were performed for each sample. The temperature dependence of the dynamic viscosity (*η*_0_) of P104/water samples was analyzed by the Andrade-Eyring equation [71]:*η*_0_ = A × e^(B/RT)^
(1)
where A and B are empirical constants in this equation. B value may represent the energy (kJ/mol) necessary for the fluid to start to flow.

### 2.4. Density and Ultrasound Velocity Measurements

Density and ultrasound velocity measurements were simultaneously, automatically, and continuously measured using an Anton Paar DSA 5000 densimeter and a sound velocity analyzer. The equipment consists of two cells for density measurements and another for the speed of sound. The temperature constant was maintained within ±1 × 10^−3^ K using the Peltier method. Density and ultrasound measurements reproducibility are ±1 × 10^−6^ g/cm^3^ and ±1 × 10^−2^ m/s, respectively. Density and sound velocity are measured using the vibrating tube method. P104/water solutions in a concentration range from 1 × 10^−4^ to 15 wt.% were placed in a venoclysis-type syringe and were subsequently injected into a chamber inside the hydrometer, letting the solution rest for 10 min to eliminate all bubbles. Experiments were programmed to perform measurements in a temperature range of 5 to 60 °C. At the end of each experiment the hydrometer chamber was washed with alconox solution, to clean all residues and avoid errors in subsequent measurements. The densimeter was calibrated with HPLC water.

The experimental measurements of density and sound velocity are used to calculate the apparent molar volume (*V_ϕ_*) and the apparent molar adiabatic compression (*K_ϕ_*), which were determined by using the experimental values obtained for P104 solutions density and sound velocity. Equations (2) and (3) were used to calculate *V_ϕ_* and *K_ϕ_*, respectively [72]. Apparent (molar) properties are not constants, even at a given temperature, but are functions of the composition.
*V_ϕ_* = [(*M_w_/ρ_s_*) − *((ρ_s_* − *ρ_w_*) × 10^3^)/(*m* × *ρ_s_* × *ρ_w_*)](2)
*K_ϕ_* = [((*β_s_* − *β_w_*) × 10^3^/(*m* × *ρ_w_*)) + (*β_s_* × *V_ϕ_*)](3)
where *ρ_s_* is the solution density (g/cm^3^) at the corresponding molality *m*, *ρ_w_* is the solvent density, and *M_w_* is the molecular weight of the solute (g/mol); *β_s_* and *β_w_* are the adiabatic compressibilities of P104 solution and the solvent, respectively, and are given by Equations (4) and (5).
*β_s_* = [(10^−3^)/(*U*^2^*_s_* × *ρ_s_*)](4)
*β_w_* = [(10^−3^)/(*U*^2^*_w_* × *ρ_w_*)](5)
where *U_s_* and *U_w_* correspond to the sound velocities of the solution and solvent, respectively, in m/s.

From density and ultrasound velocity measurements, the hydration number was determined by using the following equation:*n_H_* = lim_(*nd* → 0)_ (*n_w_/n_d_*) × (1 − *β_s_*/*β_w_*)(6)
where *n_H_* is the hydration number, *n_w_* and *n_d_* are the moles of water and moles of solute, respectively, *β_s_* and *β_w_* are the adiabatic compressibility of the solution and of pure water, respectively [73].

### 2.5. Dynamic Light Scattering Measurements

Dynamic light scattering (DLS) measurements were carried on in a Malvern Zetasizer 5000 instrument equipped with a 7132 multibit correlator and multiangle goniometer (Malvern Panalytical, Malvern, UK). The light source was a laser of He–Ne (5 mW) having a wavelength of 632.8 nm. The light-scattering intensity of P104 samples was measured through a 400 μm pinhole. The correlation functions were averaged over 60 s for 5 measurements in the equilibrated sample. DLS measurements for P104 were performed at the following angles: 45°, 90°, and 135°, in the temperature range from 10 up to 64 °C.

### 2.6. Turbidity Measurements

Temperature variations of the turbidity of the amphiphilic block copolymer P104 solutions were monitored by employing a Turbidity Measuring Module Haze QC ME from Anton Paar using a wavelength of 650 nm ± 30 nm (MEBAK- and EBC-compliant) and cell with adjustable constant temperature; measurements were performed in the temperature range from 5 to 40 °C, stabilizing the temperature with an accuracy of 0.001 °C.

### 2.7. Rheometry

The rheological behavior of P104/water system was studied using a rheometer AR-G2 from TA Instruments (New Castle, DE, USA). Two different geometries were selected depending on P104 solution concentration: (i) for P104 solutions with concentrations between 1 and 25 wt.%, a titanium cone with diameter of 60 mm and an angle of 1° was used; (ii) for P104 solutions with concentrations higher than 25 wt.% and up to 60 wt.%, a steel cone geometry with a diameter of 40 mm and an angle of 2° was used. Both geometries were used with a humidity controlling chamber to avoid evaporation of the samples.

Strain sweeps: in order to obtain the linear viscoelastic regimes, the oscillation strain sweeps were performed at a controlled angular frequency of 10 rad/s in a strain range between 0.1% and 100%, using 10 points per decade. A strain sweep was performed for each P104 sample at a selected temperature depending on the chosen concentration.Temperature sweeps: these sweeps were performed for each P104 sample using a strain value in the linear viscoelastic region, selected from previous experiments, applying an angular frequency of 10 rad/s in a temperature range from 1 to 90 °C, with a heating rate of 1 °C/min.

## 3. Results and Discussion

### 3.1. Visual Observations

Since P104 is quite soluble in water, we monitored the visual changes exhibited by the samples at different temperatures and triblock concentrations. For concentrations and temperatures lower than 20 wt.%, and 60 °C respectively, all the samples are transparent and exhibit a liquid-like behavior. At temperatures around 60 °C, i.e., 3 wt.% sample (see Appendix A) becomes bluish and slightly whitish and exhibits birefringence when it was shaken and observed through crossed polarized plates. This transition temperature can be associated with the onset of the micellar growth temperature (MGT) were the spherical-to-rod micellar transition appears. As soon as the sample temperature increases, the whiteness increases up to about 70 °C, when a separation phase is detected. This transition may be due the onset of the cloud point temperature (CPT) [70,74]. At higher concentrations, i.e., sample of 20 wt.% it exhibits a similar trend to the more diluted sample (see Appendix A), but some differences were detected; about 60 °C where the MGT appears the sample exhibits a gel-like behavior. This may be due to the growth of the rods, which forms worm-like micelles with very long length; moreover, the birefringence induced by shear becomes stronger, and the CPT moves at higher temperatures (75 °C). On the other hand, for samples more concentrated (see Appendix A), i.e., 50 wt.% the sample turned whitish, and the gel-like behavior was detected at about 50 °C; the whitish intensity decreases as temperature increases up to about a temperature of 70 °C when a second white phase appears due the onset of the cloud point temperature. In this temperature interval (50–70 °C), the sample exhibits static birefringence when it is observed through crossed polarized plates. In order to identify the microstructure, the sample was analyzed by polarized light microscopy; the textures observed are typical of the hexagonal phase (see Appendix A) [53,75,76].

### 3.2. P104 Micellization in Water Evaluated through Viscosity Measurements

The effect of temperature on the viscosity (*η*) of P104 samples was analyzed in order to detect and validate previously observed phase transitions and structural changes determined by visual observations. Measurements were performed at three different angles; viscosity data obtained were independent of the angle value used (shear rate), this means that at this concentration and temperatures range studied, samples exhibit a Newtonian behavior. Appendix A) presents the obtained results for the viscosity of P104 solutions as a function of the temperature at the following C_P104_: 1, 2, 3, 5, 8, 10, 15 and 20 wt.%. As expected, an increase in viscosity is observed with the increase on P104 concentration [77]. On the other hand, it is evident that log *η* decreases linearly as temperature increases up to a temperature interval where viscosity curves exhibit an inflexion, and then at higher temperatures log *η* decreases linearly again. In this Appendix A), one can observe that the inflection slope exhibits a change from negative to positive at concentrations higher than 10 wt.%. Figure 1 shows the log *η* as a function of the reciprocal of absolute temperature for the same samples. It can be observed that all curves exhibit three zones. In the first (I) and third (III) zones, which were detected at lower and higher temperatures respectively, data follow the Andrade-Eyring equation. The parameters A and B of Equation (1) were fitted to the viscosity data for each P104 concentration using least squares regression. Values of A and B parameters are shown in Table 1. The temperature value where the inflexion starts (I–II) moves at lower temperatures as surfactant concentration increases; these critical temperatures (T_C1_) are related to the onset of formation of spherical micelles when unimers start to aggregate and form spherical micelles (CMT) due to the increase of temperature and the dehydration of PPO segment. In contrast, the ending temperature values (II–III) of the second zone remains constant (~34 °C); these temperatures (T_C2_) are related to the end of the micellization [72,78]. Furthermore, the interval of temperatures ΔT (=T_C2_ − T_C1_) increases as triblock copolymer concentration increases; in this ΔT spherical and monomers coexist. T_C1_ and T_C2_ values for each concentration are shown in Table 2. It is possible that the inflection in viscosity is due to two contrary overlapping temperature effects: the first one the diminishing of viscosity due the temperature increment and second one the increasing viscosity due to the formation of spherical micelles induced by temperature (T ≥ CMT). For P104 concentrations ≤ 10 wt.%, it seems to be that the first effect is higher than the second one. For higher concentrations, it is evident that viscosity augments as temperature increases; in this case the increment in temperature and concentration causes an increment in the number of micelles, which produces an increasing viscosity that is larger than that the decreasing of viscosity due the augment in temperature. This effect can be seen in Table 1 where values for B_I_ and B_III_ are shown. It is clear that B_I_ is larger than B_III_ for samples with C_P104_ ≤ 10 wt.%; at higher concentrations, B_III_ values are bigger than B_I_.

The dependence of log B_I_ and log B_III_ as a function of log C_P104_ are shown in inset in Figure 1. Here it can be seen that B_I_, obtained for temperatures lower than T_C1_ (CMT), exhibits a linear dependence with C_P104_ with a power of 0.17 ± 0.02 (kJ/mol). On the other hand, B_III_, obtained at temperatures higher than T_C2_, the curve shows a change in the slope at a concentration of about 9 wt.% from 0.14 ± 0.03 to 0.77 ± 0.12. This may be due to the increase in the concentration of micelles and the interaction between micelles corona begins, which produces a greater increase in viscosity, and the beginning of the gel like behavior.

### 3.3. P104 Micellization Process Studied through Density and Sound Velocity Measurements

The study of the dependence of density and sound velocity with concentration and temperature gives information about structural changes in triblock copolymers aqueous solutions. With the obtained data, it becomes possible to determine critical concentrations, the regions where only unimers of amphiphilic block copolymers exist, the formation of spherical micelles (CMT) as well as their maximum formation rate, the sphere-to-rod-like micelle transition (GMT), and the cloud point temperature (CMT) [72].

Figure 2a shows the behavior of P104 density (*ρ_s_*) as a function of temperature for several C_P104_. In this figure it is possible to observe that density increases as P104 concentration augments and decreases with temperature up to a critical temperature value, where a more pronounced decrease is evident. Furthermore, this critical temperature shifts to lower temperature while increasing P104 concentration. A second critical temperature is detected, where density starts decreasing monotonically with temperature. These critical temperatures values are in good agreement with those detected by dynamic viscosity measurements, which correspond to T_C1_ and T_C2_, respectively. This behavior was reported elsewhere for P_103_/water [72] and P_94_/water [78] systems. This transition is then attributed to the critical micellar temperature (CMT). T_C2_ on the other hand, as was described in viscosity measurements, indicates the finishing of the spherical micelle formation, this temperature transition moves to lower temperatures as concentration increases [72,78]. This transition is 1 °C lower than that obtained with viscosity measurements. In this temperature interval (T_C1_–T_C2_), P_104_ monomers and micelles coexist, besides (T_C1_–T_C2_) becomes broader as P104 concentration decreases. At higher temperatures, a third transition is detected (T_C3_), which coincides with the growing micellar transition (GMT), which was detected by visual observations. This means that in this temperature interval (T_C2_–T_C3_) samples are formed only by spherical micelles [72].

Figure 2b depicts the dependence of P104 sound velocity (*U_s_*) with temperature for several P104 concentrations. It is evident in this figure that at low temperatures for all samples the sound velocity increases as temperature increases, the same trend as that of water, and then a transition is detected as was observed in density measurements, such as in density measurements. Sound velocity data exhibit three critical temperatures, T_C1_, T_C2_, and T_C3_, both T_C1_ and T_C2_ shift to lower temperature values as P104 concentration increases and is nearly the same as the value determined through density measurements (see Table 1). On the one hand, T_C3_ is independent of P104 concentration. Moreover, these transition temperatures are sharper than that detected by density measurements, this is due to the fact that sound velocity measurements are more sensitive at structural transitions [72]. At temperatures lower than T_C1_, the presence of P104 unimers in the solution increases with P104 concentration, so sound velocity increases also. However, at the onset of micellization, P104 unimers begin to aggregate due to the increase of temperature and the dehydration around the PPO segments. This phenomenon results in a drop of sound velocity produced by the diminution on the number of effective particles in the solution [72]. This transition is associated with the critical micellar transition (CMT) [72]. At temperatures below the micellization boundary (CMT), amphiphilic block copolymers exist as individual molecules in solution, i.e., unimers. Then, beyond the micellization border, micelles coexist in equilibrium with unimers [55]. The obtained values are in good agreement with other literature reports [2]. After the transition temperature, since C_P104_ and temperature increase, the size of the micellar aggregates also growths, resulting in stronger interactions among them and in the decrease in their number density, producing the decrease of the sound velocity values. This causes the sound velocity curves for all P104 concentrations studied to cross the water curve and then become lower than that of water at a temperature about 40 °C [72]. The *U_w_*-*U_s_* difference increases as P104 and temperature augments. In order to detect the transition temperatures values, derivatives of *ρ_s_* and *U_s_* were obtained.

Figure 3a,b, depict −*dρ_s_/dT*) and −(*dUs/dT*) respectively as a function of T. In the one hand, plot of −(*dρ_s_/dT*) against temperature shows a sharp peak between T_C1_ and T_C2_; it is evident that the peak moves at lower temperatures with increasing P104 concentration, indicating that the temperature of the onset of formation of micelles CMC (T_C1_) moves at lower temperatures as was observed by dynamic viscosity measurements; moreover, the GMT (T_C3_) is clearer in Figure 3a than that in Figure 2a. The peaks may be representing the rate of spherical micelles formation and then the maximum represents the highest rate micellar formation. On the other hand, −(*dUs/dT*) versus temperature in Figure 3b, depicts a similar trend to −*dρ_s_/dT* against temperature. The three critical temperatures are detected; however, the critical temperature T_C3_ is more evident in this case.

Then, Equation (3) was used to calculate the apparent molar adiabatic compression (*K_ϕ_*), which is directly related to the apparent molar volume (*V_ϕ_*). The apparent molar adiabatic compressibility (*K_ϕ_*) was inspected as a function of temperature for P104 solutions with different concentrations (see Appendix A). Two linear regions with a sharp increase with temperature are identified in all the curves. First onset is directly related to the CMT, as previously described. First derivative of *K_ϕ_* was then determined as a function of temperature (see Appendix A). A sharp peak is detected for each C_P104_ and is directly related to the onset of micellization, the CMT. As expected, a shift of the position of the peaks to lower values of temperature is observed with the increase on P104 concentration. At higher temperatures, a slight shoulder is detected around 55 °C, which could be related to a transition from micelles to rod-like micelles (GMT). However, since micellization of P104 block copolymer takes place at higher temperatures than other triblock copolymers [1,2,4,72] the use of techniques such as density measurements is limited by the temperature interval measurements of the equipment. The CMT values obtained by both density and sound velocity measurements coincide (see Table 2).

### 3.4. Evaluation of the Hydration Number of P104/Water System

In previous reports in the literature [78], *V_ϕ_* and *K_ϕ_* data obtained from aqueous solutions of some drug compounds have been qualitatively interpreted in terms of solute–solvent and solute-solute interactions. There, the adiabatic compressibility was correlated with the hydrational behavior of the solute molecule and was found to be sensitive to the structural features of the solute, such as shape, size, branching, and presence of aromatic rings. Furthermore, it has been proposed that relationships between the chemical structures and hydration environment of polymers can provide significant comprehension of water–amphiphilic polymer interactions [79].

Figure 4a shows the hydration number of P104/water system obtained using Equations (4)–(6) as a function of temperature for a set of P104 concentrations (from 1 to 20 wt.%). It can be observed that, independently of the temperature, hydration number decreases while P104 concentration increases. A similar trend was detected by Liu et al. for P84 and P104 triblock copolymers aqueous solutions performed by Small Angle Neutron Scattering (SANS) [60]. They reported that, at given temperature, the number of water molecules bounded is independent of triblock concentration. The micelles formed with P84 copolymer are less hydrated than those formed with the P104 copolymer; the hydration number decreases, and the micelles become more compact as temperature increases. Here, it has been mentioned previously, the dehydration around the hydrophobic PPO segments forming the core and the hydration of hydrophilic PEO segments forming the corona are responsible for amphiphilic block copolymer micelle formation [72,78].

Figure 4a shows, as an example, an inflection points in the hydration number curve at the temperature of 20 °C for the P104/water solution having a concentration of 8 wt.%. The inflection point; attributed to the critical micellar temperature (CMT); decreases as the concentration of the system increases. A decrease in CMT with an increase in amphiphilic block copolymer concentration has been widely observed and is a well-known effect reported for several Pluronic systems [2,55,80,81,82]. The CMT values for the Pluronic copolymer solutions (at a given copolymer concentration) decrease also as a function of the number of PO segments, showing that polymers with a larger hydrophobic domain form micelles at lower temperatures [2]. At temperatures higher than 47 °C, the hydration number exhibits negative values. This is due to the crossover of the sound velocity curves for all P104 with the water curve, where they become lower than that of water (see Figure 3b). In order to obtain accurate critical temperatures, the derivative of the hydration number as a function of temperature was calculated, and the curves obtained are depicted in Figure 4b. It can be seen in this figure that all samples show a sharp peak; which, as was mentioned, is related to the onset of micellization (CMT) and the maxima in the peaks represent the maximum rate velocity of dehydration and therefore the maximum rate of micellization; the dehydration rate exhibits an inflection with P104 concentration at around 9 wt.% similar to that observed in viscosity measurements. Moreover, peaks exhibit a shift of the position of the peaks to lower values of temperature with increasing P104 concentration. At higher temperatures, a third transition is detected at about 55 °C, which may be related to a transition from micelles to rod-like micelles. The CMT and GMT values obtained by hydration measurements are depicted in Table 2. The hydration number obtained by density and sound velocity in this work are lower to that reported by Liu et al. [60].

Furthermore, when plotting the hydration number as a function of P104 molar concentration (Figure 5a), it is possible to see that it decreases when either P104 concentration or temperature increases. The fit to the experimental data was performed in order to obtain the hydration number values at infinite dilution (*n_H_*_0_). The ln *n_H_*_0_ as a function of 1/T is shown in Figure 5b. In this Figure, four zones that are bounded by three critical temperatures can be observed in this Figure. In the zone I, *n_H_*_0_ decreases up to around a temperature of 26 °C, where the curve presents a slope change, this transition may be associated with the onset of the CMT. The second zone (II) was detected in the temperature interval from the CMT to 40 °C; this second temperature transition may be related to the finishing of the monomer–spherical micelle transition. The last temperature transition appears at 52 °C, which coincides with the shape change of micelles from spherical to prolate. Finally, from Figure 5b, the slopes for the three zones where ln(*n_H_*) are plotted as functions of the reciprocal of the absolute temperature, and were adjusted with a linear regression, An equation of the following form can be written
ln *n_H_*_0_ = A − B × (1/T)(7)
where B is the slope and A is the intercept with the *y*-axis. In analogy with the van’t Hoff equation for a chemical reaction [83], it was proposed that B should be ΔE_DH_/R, where ΔE_DH_ is the driving dehydration energy and R is the gas constant; ΔE_DH_ values for the three zones are shown in Table 3.

### 3.5. Morphology of P104 Micelles in Water by Dynamic Light Scattering (DLS)

Information about the shape of the micelles was obtained, as first approximation, by performing Dynamic Light Scattering (DLS) measurements in a temperature range from 10 to 64 °C, from the combination of the scattering intensity and the hydrodynamic radius [84,85,86,87]. The information about the hydrodynamic radius of particles is usually obtained through DLS measurements by using the Stokes–Einstein equation (Equation (8)) and the measured diffusion coefficient (D). For these experiments, low concentrations of copolymer need to be selected so concentration will not affect the diffusion coefficient. Here, it was noticed that between 0.5 and 2 wt.%, reliable data were still obtained. Above these concentrations, an apparent diffusion coefficient and a smaller apparent hydrodynamic radius was obtained.
*R_h_* = (*k_B_T*)/(6π*η_s_*D)(8)
where, *k_B_T* is the Boltzmann constant, *η_s_* is the viscosity of the solvent, and *T* is the absolute temperature.

Figure 6 shows the effect of temperature on the hydrodynamic radius, *R_h_*, and the scattered light intensity (I_SCA_) for a 1 wt.% P104 in water. From these results it was possible to identify that micelles are formed in a temperature range between 26 and 54 °C. Their average hydrodynamic radius is 11.4 ± 1 nm. The low scattered light intensity observed at temperatures below 26 °C is directly related to the presence of P104 unimers in the solution [60]. Therefore, the temperature of 26 °C is related to the CMT of P104 triblock copolymer at 1 wt.%, which is in good agreement with the CMT value reported in the literature for a P104/water system evaluated at the same concentration [2,11]. After the appearance of the CMT, while increasing the temperature, the scattered light intensity also increases and remains almost constant in the following temperature range: from 26 to 54 °C, though, after 55 °C, the progressive increase of I_SCA_ until reaching a two-magnitude order difference from the initial value suggests the appearance of a new P104 structure. This morphology transition, due to an increase in temperature, is in good agreement with the one obtained through the evaluation of *dK_ϕ_/dT*. This phenomenon has been described in terms of the enhanced dehydration of the micelle corona, consisting essentially of PEO, during the increase of temperature [55,88].

Furthermore, the formation of new structures was proposed through the analysis of the variations of the intrinsic asymmetry [Z] [89]. This parameter was obtained by calculating the ratio of the scattering intensity values measured at 45° and the scattering intensity values measured at 135° (I_45°_/I_135°_) (see Appendix A). The analysis of this aspect factor I_45°_/I_135°_ as a function of temperature for a P104/water solution having a concentration of 1 wt.% allowed identifying the onset of an I_45°_/I_135°_ ratio different than 1 at the temperature of 54 °C. Then, the ratio between the characteristic dimension of the structure and the wavelength (D/λ) is found to be around 0.04 when [Z] ≈ 1, i.e., between 10 and 54 °C, suggesting spherical micelle morphology [89]. Micellar growth is then detected at higher temperatures.

Furthermore, another approximation about micelle shape was determined through the Perrin model [90]. This model is used to estimate the sizes of micelles for prolate and oblate ellipsoids [91,92,93] The expression for the prolate case is given by the following equation:*R_h_* = (*b*/2) × [(*p*^2^ − 1)^1/2^/(ln(*P* + (*p*^2^ − 1)^1/2^))](9)
where *p = a/b*, *b* corresponds to the semiminor axis and a corresponds to the semimajor axis. For prolate, a is the micelle length L and *b* corresponds to the diameter of the spherical micelle *b = 2R_h_*^0^. In this case, *R_h_*^0^ corresponds to the hydrodynamic radius of the micelle at 38 °C.

On the other side, the expression for the oblate ellipsoid is given by the following equation:*R_h_* = (*a*/2) × [((1*/p*)^2^ − 1)^1/2^/(arctan((1*/p*)^2^ − 1)^1/2^)](10)
where *a = 2R_h_*^0^.

The volume of the micelle (V_mic_) is given by the following expression:I_SCA_ αV_mic_
*× P(q)*(11)
where I_SCA_ is the total scattering intensity and *P(q)* is the micelle form factor. *P(q)* for spherical, prolate, and oblate ellipsoids can be determined by using the Debye and Anacker equation [93].

The scattered intensity I_SCA_/I_SCA_^0^ as a function of the hydrodynamic radius *R_h_* is presented in Figure 7 for a 1 wt.% P104 solution in water. The curve corresponding to I_SCA_/I_SCA_^0^ is compared to the Perrin model of prolate ellipsoids, oblate ellipsoids, and spheres. I_SCA_ and R_h_ were measured at six different temperatures from the spherical micelle domain to the elongated micelles domain. I_SCA_^0^ is the scattering intensity taken at 38 °C, which corresponds to the initial temperature, selected in the spherical micelle domain. The evolution of I_SCA_/I_SCA_^0^ as a function of *R_h_* is close to the predicted behavior for prolate ellipsoids, suggesting that P104 micelles grow as prolate rods. A similar behavior was also obtained for P103 micelles in water [90].

A spherical micelle morphology of P104 triblock copolymer can be proposed from this DLS study since I_SCA_ did not show any dependence on the scattering angle. Above 54 °C, both I_SCA_ and *R_h_* increase steadily with increasing temperature, suggesting a structural transition from spheres to elongated micelles. This conclusion is supported by the variation of the aspect factor (I_45°_/I_135°_) from ~1 between 25 and 54 °C to above 1.5 for T > 54 °C (see Appendix A). Then, the evolution of I_SCA_ as a function of *R_h_* was also compared with the Perrin model, taking *b =* 2*R_h_*^0^ and *a =* 2*R_h_*^0^ for prolate ellipsoid and oblate ellipsoid respectively, determining a small prolate rod growth.

The obtained micellar structure for P104 amphiphilic copolymer in water in the temperature range between 25 and 55 °C, corresponding to the spherical micelle domain, is suitable to be used a micellar nanocarrier for drug-controlled release and their dynamics are now being studied with lipidic membranes through hydrophobic pyrene probe transfer [11]. Their nanoscale size makes them a suitable option for targeted drug delivery applications, including storage, controlled release, and protection of the hydrophobic drugs [94]. Contrary to P103 triblock copolymer, which starts to aggregate at 37 °C, the spherical micellar stability of P104 at the temperature of 37 °C represents a good selection that allows maintaining a specific shape and size at the average body temperature. Then, the morphology could be modified through a determined stimulation (temperature, pH or ionic strength variations) in order to release the hydrophobic drug. The shape of micelles is affected by temperature, which can be exploited for the design of new formulations.

### 3.6. Turbidity Measurements

Turbidimetry is a powerful method that reveals the aggregation behavior in copolymer solutions that exhibit micellization temperatures [95] Figure 8a shows the dependence of turbidity with temperature and P104 concentration. All of the P104 solutions exhibit an initial slight variation of the turbidity between the temperatures of 10 to 20 °C. The higher the concentration of 10 wt.% P104 is, an increasing and then a decrease in turbidity is observed, while increasing temperature becomes more important. This behavior was also observed in methoxy-poly-(ethylene glycol)-block-poly(N-isopropylacrylamide) (MPEG_53_-b-P(NIPAAM)_113_ [95] diblock copolymers and was attributed to possible shrinking of the polymeric micelles before intermicellization occurred [95]. Then, a sharp rise of the turbidity observed in all P104/water solutions was attributed to the self-organization of amphiphilic block copolymer chains to form micellar structures, i.e., the boundary between unimers and unimers-spherical micelles, as mentioned before. At higher temperatures, the turbidity becomes a constant value, i.e., 40 °C for the 10 wt.% sample; this temperature coincides with the finishing of the transition of unimer to spherical micelle for this concentration. The obtained temperature values are in good agreement with the previously mentioned results from density, sound velocity, and viscosity measurements (see Table 2).

The first derivative of the turbidimetry with temperature allows also identifying the boundary between unimer and unimer–spherical micelle zones. Figure 8b shows the variation of d turbidity/dT with temperature for some P104 concentrations. A maximum is observed between 22 and 28 °C, depending on P104 concentration, and is related to the critical micellar transition (CMT). In Table 2 the critical temperatures measured are depicted. It can be seen that the values obtained are in agreement with those obtained with the other analytical methods used in this work [96]. The inset in Figure 8b shows the maximum of the derivative of the turbidity with temperature as a function of P104 concentration; a maximum is detected around 8 P104 wt.%. This value is similar to that detected by viscosimetry and hydration measurements.

### 3.7. Rheological Measurements

The ability of the hydrophobic core of PPO in amphiphilic triblock copolymers to delay the release of hydrophobic drugs is an important feature on Pluronics for drug delivery. Thermo-reversible and physical gels can be formed from some of these copolymer solutions at higher polymer concentrations and temperatures [27,97]. These gels consist of liquid crystals of packed spherical or rod-like micelles. Different properties of the materials can be studied through rheological measurements and are useful in many biomedical applications [98,99]. With this technique it is possible to measure the transitions at a temperature higher than the analytical methods discussed above.

#### 3.7.1. Linear Viscoelastic Region

P104/water samples were characterized mechanically in the linear viscoelastic region (LVR), which is defined as the deformation range where the elastic modulus (G′) is independent of deformation (% γ) [100]. After a critical deformation (% γ_c_), G′ exhibits a change in its slope, decreasing as deformation increases, which is due the breakdown of the sample microstructure. In order to identify the linear viscoelastic region, oscillatory strain deformation sweeps for P104/water solutions in a concentration range between 5 and 80 wt.% in the deformation interval from 0.01 to 100% were achieved at a frequency (ω) of 10 rad/s and at various temperatures. Figure 9a shows, as examples, the strain dependence of G′ and G″ for five P104 concentrations at the temperature of 20 °C. It is evident in this Figure that in the semi-dilute and dilute P104 concentrations (≤25 wt.%) the LVR reach deformation values up to around 100%. For more concentrated samples, the LVR decreases at deformations about 1%. It can be observed that for a P104 concentration of 5 wt.%, G′ and G″ are independent of γ at strains less than the 20%, however, for a P104 concentration of 35 wt.%, G′ and G″ are independent of γ at strains less than the 1%. This diminishing of the LVR with increasing concentration may be due the formation of a weaker structure. In Figure 9b the elastic modulus as a function of deformation for a sample of 30 wt.% at different temperatures is shown. For a temperature of 10 °C; which is below the CMT, the LVZ reaches at levels of deformation of 100%. When the deformation sweep is performed at temperatures of 30 and 60 °C that is higher than the CMT, sample becomes structured, the modulus increases in 7 and 6 magnitude orders, and the critical deformation shift at 1 and 7% respectively. Therefore, it was found that the linear viscoelastic region is highly dependent in both P104 concentration and temperature.

#### 3.7.2. Temperature Sweeps

Figure 10 presents a set of plots showing the temperature dependence on the log (G′) and log (G″) at a frequency (ω) of 10 rad/s for different P104/water solutions having the following concentrations: 5, 10, 20, and 30 wt.%. For P104 concentration of 5 wt.% (Figure 10a), a slightly elastic behavior (G″ < G′) is firstly observed within the temperature range from 3 to 19 °C where a change in the slopes is detected, this change is associated with the onset of the CMT; then the elastic and loss moduli decreases with temperature until reaching the temperature of 60 °C, at which a large increase of two orders of magnitude is observed, changing from the liquid to the soft gel domain; this transition is due to the growth of the rods, which form wormlike micelles. G′ increases with temperature up to around 75 °C becomes a maximum and then diminishes with temperature; this temperature transition is due to the cloud point temperature of the P104/water system detected by visual measurements. For P104 concentration of 10 wt.% (Figure 10b), a viscous behavior (G″ > G′) is firstly observed within the temperature range from 3 to 20 °C. A first crossover of G′ and G″ is then observed around the temperature 20 °C. The elastic modulus is then independent of the temperature until reaching the temperature of 58 °C, at which a large increase of two orders of magnitude is observed, changing from the dilute to the soft gel domain. A drop of both G′ and G″ values is then observed at the temperature of 81 °C, at which the soft gel condition returns to a liquid condition. A small pick is detected around 17 °C before the G′ and G″ crossover, this temperature is related to the onset of the formation of spheric micelles. In the other hand the large increment of both G′ and G″ is due to the growth of length of the rodlike to form wormlike micelles (GMT). The maximum detected at 75 °C is due to the onset of the cloud point temperature (CPT). Figure 10c shows the temperature dependence of G′ and G″ for the P104 concentration of 20 wt.%. A similar behavior to the one obtained for P104 concentration of 10 wt.% is observed. Here, the loss and the storage modulus decrease with temperature until reaching the temperature of 11 °C, at which a change in G′ slope is observed; at higher temperatures around 60 °C, a large increase of also two orders of magnitude is observed, changing from the liquid to the soft gel domain; which indicates the onset of GMT. The drop of both G′ and G″ values is slightly shifted to higher values of temperature, i.e., 83 °C. Finally, a new behavior of P104 solutions is presented in Figure 10d for a sample of 30 wt.%, in which the material reaches the hard gel domain between a temperature range of around 21 and 52 °C. In this manner, an increase of around four orders of magnitude is reached from this P104 concentration in a ΔT of around 30 °C. Therefore, it is possible to conclude that P104/water solutions form thermo-reversible gels depending on P104 concentrations and on temperature, which could be used for different applications at physiological temperatures [27,32,101]. Dashed horizontal lines in Figure 10 represent the liquid-like behavior (sol, with G′ < G″ and G′ < 10 Pa), the soft gel behavior (with G′ > G″ and 10 < G′ < 1000 Pa), and hard gel behavior (with G′ > G″ and G′ > 1000 Pa); these rheological behaviors and the G′ values limits were adopted from Hvidt et al.’s notation [27].

### 3.8. Temperature-Composition Phase Diagram of P104/Water in the Dilute and Semi-Dilute Regimes

Figure 11 shows a phase diagram for P104/water system. P104/water solutions having concentrations from 1 × 10^−4^ to 90 wt.% were inspected as a function of temperature (from 5 to 80 °C) to determine the following boundaries: (i) liquid-soft gel, (ii) soft gel-hard gel, (iii) not birefringent/birefringent and (iv) transparent/opaque. Rheological measurements, previously described, allowed determining P104/water solutions having concentrations that behave like liquid, soft gel, or hard gel materials [102,103,104]. Furthermore, samples that exhibited dynamical and static birefringence (observed through crossed polarizers) were examined using a crossed-light polarized microscope. Appendix A) show the optical micrographs of P104/water samples having concentrations of 50 and 30 wt.%, analyzed at the temperatures of at 80 and 74 °C, respectively. The observed textures are characteristic of hexagonal liquid crystals [75,76].

Figure 12 presents the phase diagram for P104/water system in the C_P104_ range from 0 to 20 wt.%. Here it can be observed that the CMT slightly decreases with the increase of P104 concentration, as previously reported for P104 concentrations from 0.01 to 10 wt.% [2]. Density, sound velocity, dynamic light scattering, and viscosity measurements allowed detecting the existence of micelles and unimers in a wide range of temperatures. The start and the end of the peak of the derivative of the apparent molar adiabatic compressibility (represented by the dashed line) were used to identify the CMT and the temperature at which the micellization is finished. After the GMT (micellar growth temperature), spherical micelles grow into rod-like micelles (prolates), according to the first approximation determined by using Perrin’s model. At higher temperatures, a reversible phase separation takes place and the clouding point is reached, at which a great storage modulus (G′) is detected.

## 4. Conclusions

In this article, a detailed study of amphiphilic block copolymer P104 was carried out through rheometry and density, sound velocity, viscosity, and DLS measurements in the dilute and semi-dilute regimes between 5 and 90 °C. The hydration number of P104/water system was determined with the resulting data of density and sound velocity measurements, for a set of P104 concentrations, from 1 to 15 wt.%, as a function of temperature. An inflection point in the hydration number curve as a function of temperature was attributed to the CMT, which decreases as P104 concentration increases. The dehydration around the hydrophobic PPO segments forming the core and the hydration of hydrophilic PEO segments forming the corona are responsible of amphiphilic block copolymer micelle formation.

The obtained results from viscosity, density, sound velocity, hydration, DLS, and rheometry allowed analyzing the structural behavior of P104/water system. The critical micellar temperature (CMT) and the micellar growth temperature (MGT) were determined as a function of C_P104_. It was found that the temperature domains at which P104 spherical micelles and P104 elongated micelles exist are greater than for other triblock copolymers, allowing their applications in a wider field.

The scattered intensity (I_SCA_) and the hydrodynamic radius (*R_h_*) were obtained through DLS measurements at six different temperatures from the spherical micelle domain to the elongated micelles domain. The dependence of I_SCA_/I_SCA_^0^ with *R_h_* was compared to the Perrin model of prolate ellipsoids, oblate ellipsoids, and spheres, and was found to be close to the predicted behavior for prolate ellipsoids, suggesting that P104 micelles grow as prolate rods. The obtained micellar structure for P104 in water in the temperature range between 25 and 55 °C is suitable to be used as a micellar nanocarrier for drug-controlled release. Their nanoscale size (*R_h_* of 11.4 ± 1 nm) makes them a suitable option for targeted drug delivery applications, including storage, controlled release [105,106,107], and protection of the hydrophobic drugs [108,109,110], as shown in our last study of exchange dynamics with lipidic membranes through hydrophobic pyrene probe transfer [11].

Rheological properties were studied in a P104 concentration range from 5 to 60 wt.% and were found to be greatly dependent on temperature and concentration, since the storage modulus increases between two and three orders of magnitude.

We also report a temperature-composition phase diagram of P104/water system, obtained through visual, optical microscopy and rheological measurements, by using the Hvidt et al. notation, for C_P104_ from 1 × 10^−4^ up to 90 wt.% in the temperature range from 5 to 80 °C. The following boundaries: liquid–soft gel, soft gel–hard gel, not birefringent/birefringent and transparent/opaque, were clearly identified. Finally, we propose a partial phase diagram temperature-composition of the system in the C_P104_ range from 0 to 20 wt.%, in which the appearance of the different morphologies was identified: unimers (U), spherical micelles (SM), cylindrical micelles (CM), and worm-like micelles (WM).

## Figures and Tables

**Figure 1 polymers-15-02551-f001:**
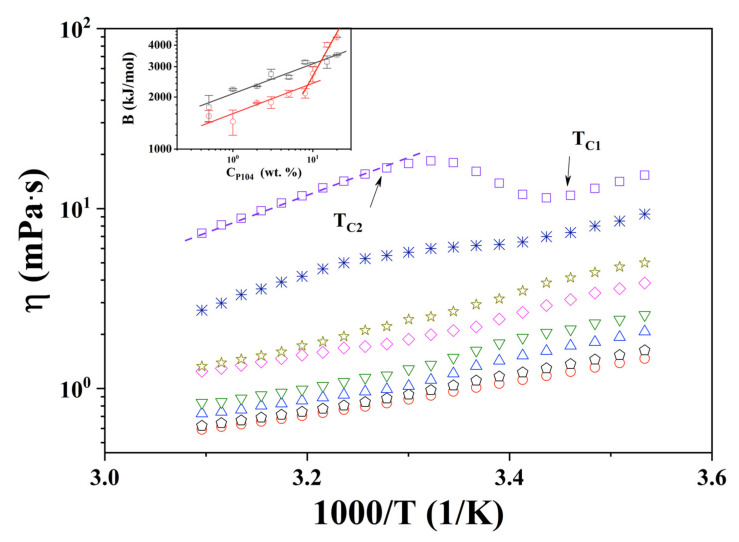
Dependence of the dynamic viscosity (log *η*) with reciprocal of absolute temperature (1/T) for P104 concentrations of 1 (◯), 2 (⬠), 3 (△), 5 (▽), 8 (♢), 10 (☆), 15 (✳) and 20 (☐) wt.%. Measurements performed at three different angles 30°, 50° and 70°. Inset: Parameter B as a function of C_P104_ for zone I (B_I_) (☐) and zone II (B_II_) (◯).

**Figure 2 polymers-15-02551-f002:**
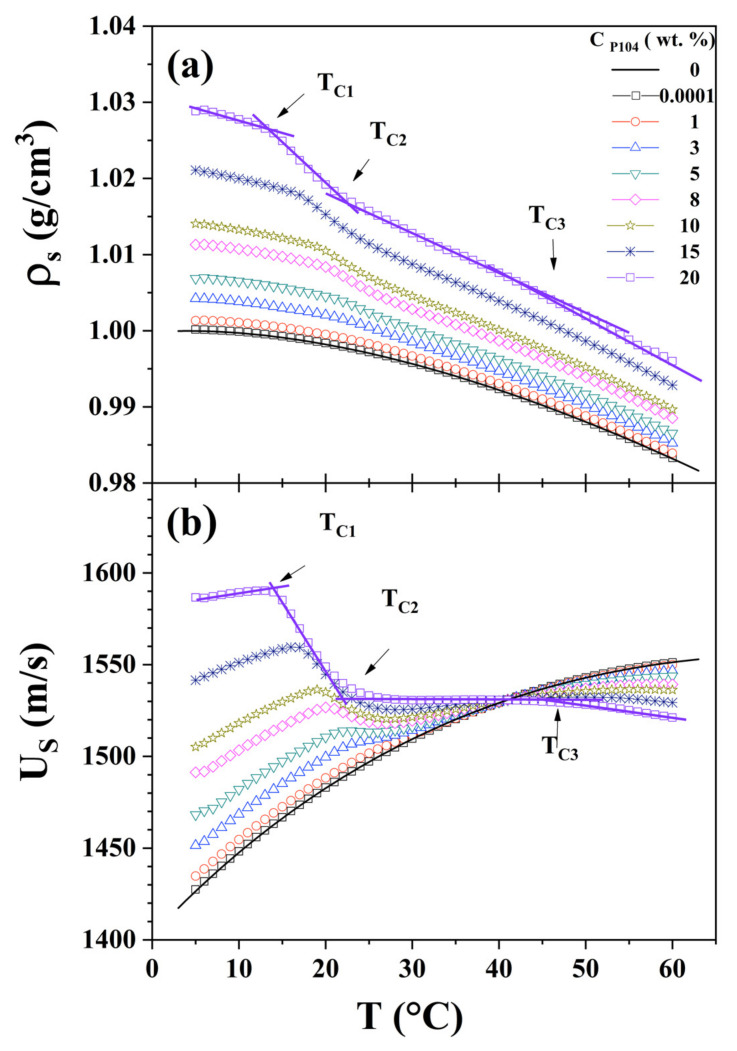
(**a**) Density (*ρ_s_*) as a function of temperature for different P104 concentrations: 0 (-), 0.0001 (☐), of 1 (◯), 2 (⬠), 3 (△), 5 (▽), 8 (♢), 10 (☆), 15 (✳) and 20 (☐) wt.% (**b**) Sound velocity (*U_s_*) as a function of temperature for the same P104 concentrations.

**Figure 3 polymers-15-02551-f003:**
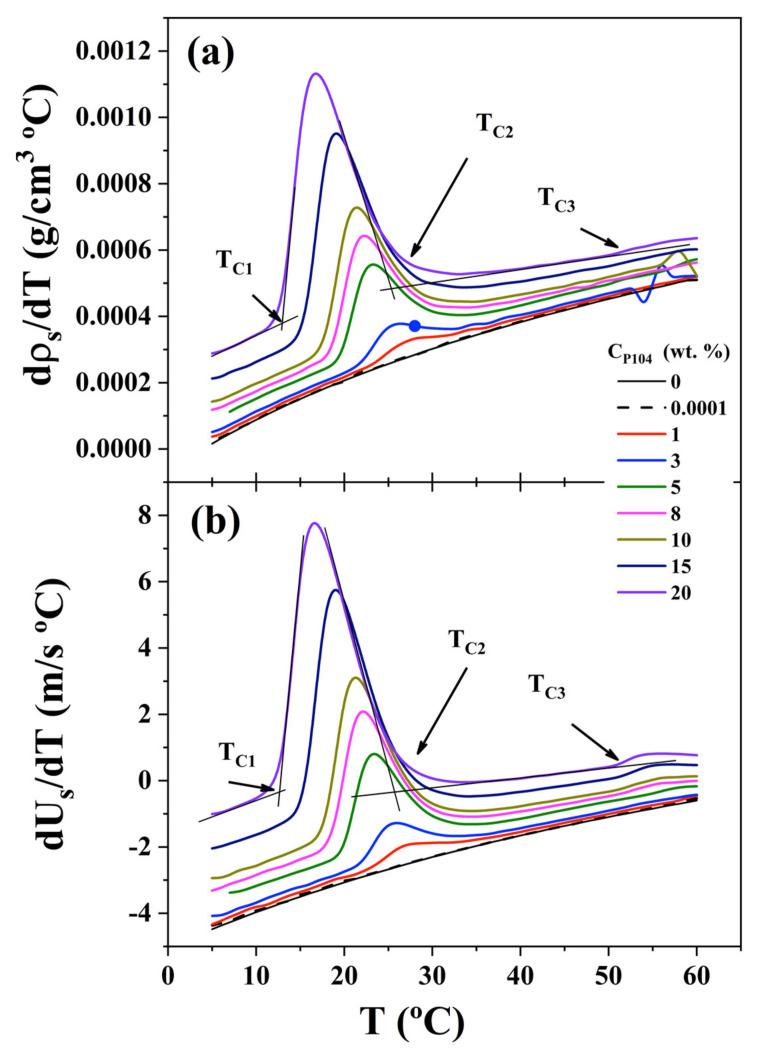
(**a**) Derivative of density (*dρ_s_*/*dT*) as a function of temperature for different P104 concentrations: 0, 1, 3, 5, 8, 10, 15, and 20 wt.%. (**b**) Derivative of sound velocity (*dUs*/*dT*) as a function of temperature for the same P104 concentrations.

**Figure 4 polymers-15-02551-f004:**
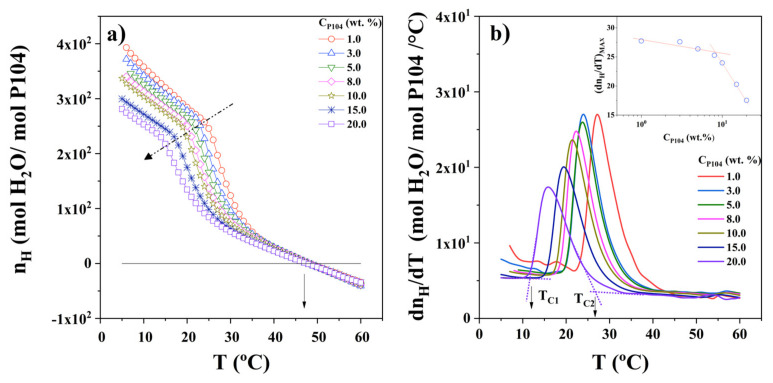
(**a**) Hydration number (*n_H_*) as a function of temperature for different P104 concentrations: 1, 3, 5, 8, 10, 15 y 20 wt.%. (**b**) Derivative of hydration number as a function of temperature for different P104 concentrations.

**Figure 5 polymers-15-02551-f005:**
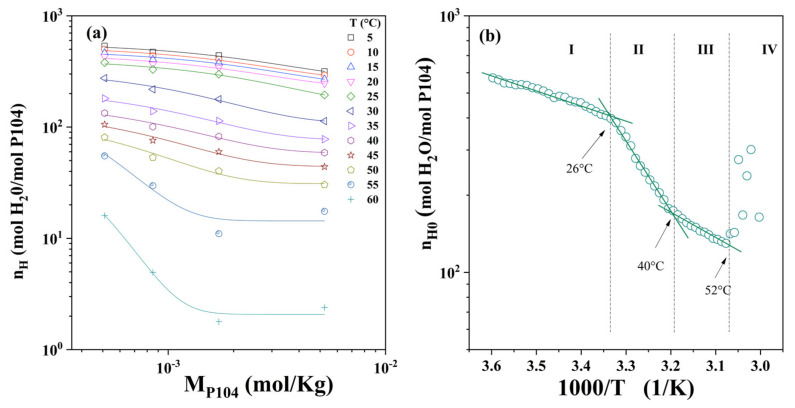
(**a**) Hydration number (*n_H_*) as a function of P104 concentration in mol/kg for different temperatures (from 5 to 60 °C). The lines correspond to no linear fit of n_H_ = ae^−(b × M*P104*)^
*+ c* (average R^2^ = 0.94 ± 0.02). (**b**) Arrhenius-type dependence of the hydration number at zero P104 concentration with the reciprocal of the absolute temperature 1/T (K^−1^).

**Figure 6 polymers-15-02551-f006:**
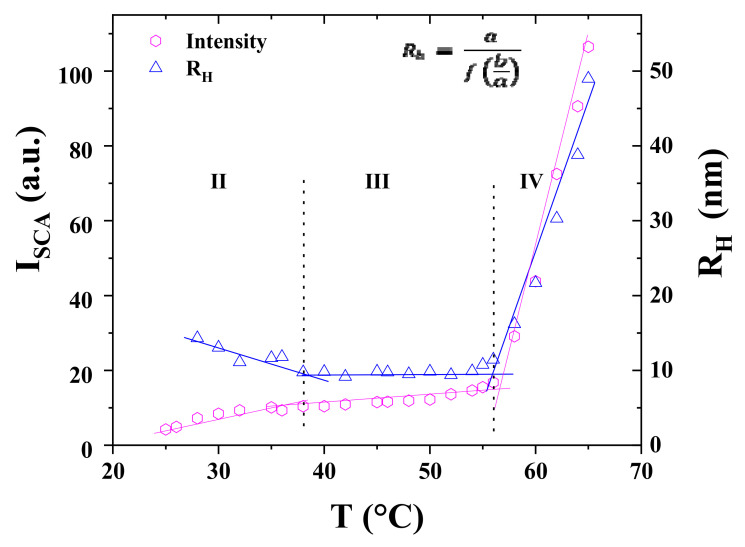
Temperature dependence of the hydrodynamic radius (*R_h_*) and the scattered light intensity (I_SCA_) for 1 wt.% P104 solution in water measured at 90°. The sample was equilibrated at the initial temperature during 24 h before each measurement.

**Figure 7 polymers-15-02551-f007:**
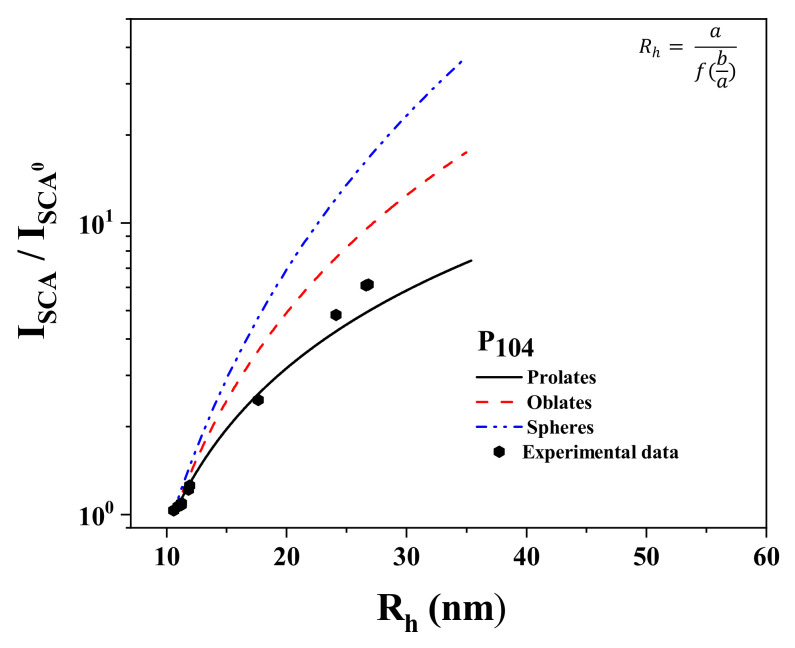
Normalized scattered intensity I_SCA_/I_SCA_^0^ plotted as a function of the hydrodynamic radius *R_h_* for a P104 solution in water with a concentration of a 1 wt.%. I_SCA_ and *R_h_* were measured at different temperatures. ISCA0 corresponds to the scattering intensity at 38 °C. The plot I_SCA_/I_SCA_^0^ is compared to the Perrin model of prolate ellipsoids, oblate ellipsoids, and spheres.

**Figure 8 polymers-15-02551-f008:**
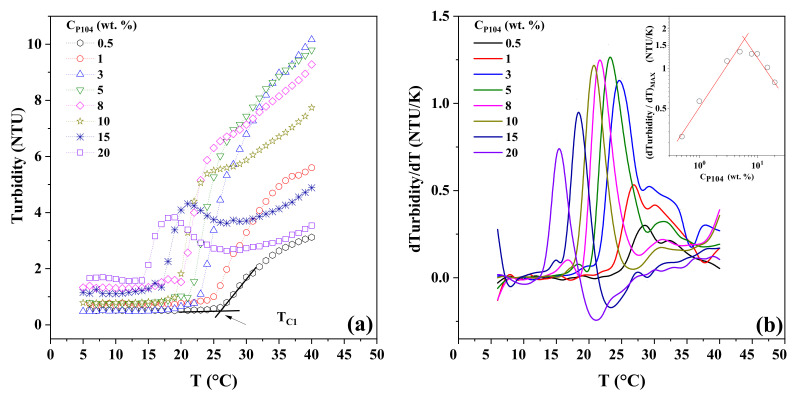
(**a**) Temperature dependences of the turbidity during heating for P104/water system as a function of P104 concentration. (**b**) Dependence of the first derivative of turbidity (d_turbidity_/dT) with temperature for different P104 concentrations.

**Figure 9 polymers-15-02551-f009:**
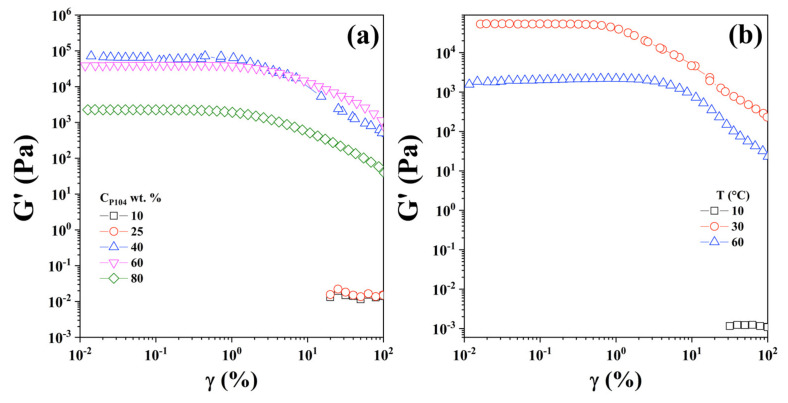
Strain dependence of G′ and G″ for (**a**) P104 concentrations of 10, 25, 40, 60, and 80 wt.% at the temperature of 20 °C at a frequency of 10 rad/s and for (**b**) P104 concentration of 30 wt.% at 10 rad/s at temperatures of 10, 30, and 60 °C.

**Figure 10 polymers-15-02551-f010:**
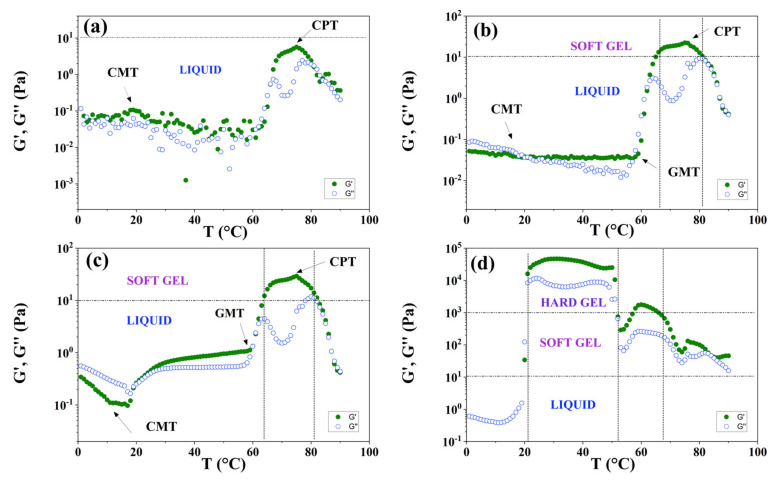
Temperature sweeps of P104/water solutions at the following P104 concentrations: (**a**) 5, (**b**) 10, (**c**) 20, and (**d**) 30 wt.%. Green circles correspond to G′ and blue circles correspond to G″.

**Figure 11 polymers-15-02551-f011:**
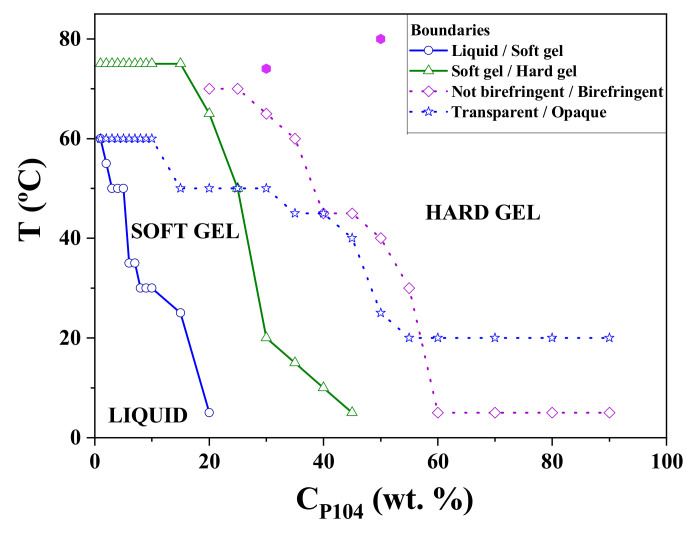
Temperature-composition phase diagram of P104/water system, obtained by visual, optical microscopy, and rheological measurements (using the Hvidt et al. notation [27]).

**Figure 12 polymers-15-02551-f012:**
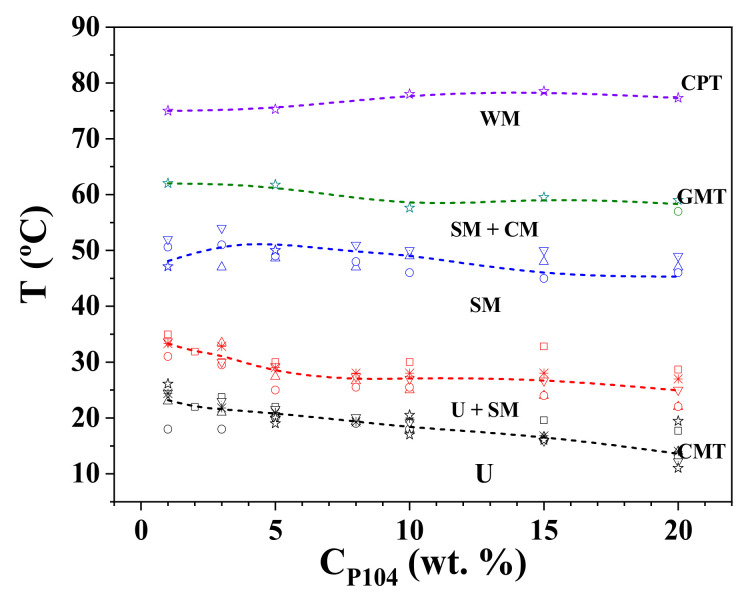
Temperature–composition partial phase diagram of P104/water system in the P104 concentration range from 0 to 20 wt.%, obtained by various analytical methods: viscosimetry (☐), density (◯), sound velocity (△), hydration number (▽), turbidity (✳), rheometry (☆), (----- statistical average), unimers (U), spherical micelles (SM), cylindrical micelles (CM), worm-like micelles (WM).

**Table 1 polymers-15-02551-t001:** Parameters A and B of Equation (1) obtained by fitting dynamic viscosity experimental data. A and B correspond to the empirical constants in Equation (1). B value represents the energy (kJ/mol) necessary for the fluid to start to flow.

C_P104_ (wt.%)	A_I_ × 10^4^ (mPa·s)	B_I_ (kJ/mol)	A_III_ × 10^4^ (mPa·s)	B_III_ (kJ/mol)
1	6.20 ± 0.4	2213 ± 24	47.80 ± 4.2	1443 ± 240
2	4.60 ± 0.2	2309 ± 12	19.90 ± 0.8	1852 ± 12
3	2.30 ± 0.2	2718 ± 180	23.30 ± 8.8	1864 ± 144
5	2.60 ± 0.4	2610 ± 60	13.90 ± 1.9	2092 ± 96
8	0.51 ± 0.14	3187 ± 60	25.80 ± 4.3	2104 ± 132
10	0.79 ± 0.01	3079 ± 84	1.60 ± 0.8	2730 ± 276
15	0.46 ± 0.01	3199 ± 265	0.12 ± 0.06	4017 ± 120
20	0.62 ± 0.05	3512 ± 24	0.08 ± 1 × 10^−4^	4438 ± 12

**Table 2 polymers-15-02551-t002:** CMT, MGT, and CPT obtained for P104/water system as a function of P104 concentration using different analytical techniques: viscosity, density, sound velocity and rheometry.

CMT (°C)							MGT (°C)			CPT (°C)	
C_P104_ (wt.%)	*η*	*dρ_s_/dT*	*dUs/dT*	*dn_H_/dt*	*Turb.*	Rheometry	*dρ_s_/dT*	*dUs/dT*	*dnH/dt*	Rheometry	
1	24	24	22	25	24.5	24	-	-	51	62	75
3	21	22	21.5	23.5	22.5	-	-	-	50	-	-
5	20	20	20	19.5	21.4	19				60	75
7	-	19	19	19	-	-					
8	19	18.5	18.5	20	20	-	53	51	51	--	--
10	18	17.6	17.5	19	19	17	56	51.5	51	58	75
15	16	15.3	15.2	16.5	16.7	16	51	51.5	50	59	76
20	15	13	12.8	14	14.3	11	51	52	50	59	75

**Table 3 polymers-15-02551-t003:** Intercept (A), dehydration energy (ΔE_DH_) and R^2^.

Region	exp (A)	ΔE_DH_ (KJ/mol)	R^2^
I	5.4500 ± 0.86	10.8 ± 0.4	0.9741
II	1.6700 × 10^−7^ ± 8 × 10^−8^	53.9 ± 1.2	0.9944
III	0.055 ± 0.014	20.9 ± 0.7	0.9862

## Data Availability

Data can be found in the figures from the manuscript.

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
