# Peer review of "Structural Behavior of Amphiphilic Triblock Copolymer P104/Water System"

_polymers, 2023, doi:10.3390/polym15112551_

Round 1
Reviewer 1 Report
This manuscript describes the structural transitions of triblock copolymer P104 in water covering semidilute to dilute regions as a function of system temperature and polymer concentration. The investigation has been carried out using viscometry, DLS, turbidimetry, microscopy and rheometry. The authors have identified the regions where monomers exist, spherical micelle formation, elongated cylindrical micelles formation, clouding points, and liquid crystalline behavior. The boundaries: liquid-soft gel, soft gel-hard gel, not birefringent/bi-refringent and transparent/opaque, are identified and finally a partial phase diagram temperature-composition of the system in the CP104 range from 0 to 20 wt.%, in which the appearance of the different morphologies was identified: unimers, spherical micelles, cylindrical micelles, worm like micelles. This work seems interesting, but the following minor issues are to be rectified as commented below before publication.
· Authors should revise typographical and grammatical errors with better description of unclear sentences for easy understanding.
· Page 6, ηo, the o should be as subscript. The equation (6) also needs to be corrected accordingly.
Author Response
Answers to reviewer 1:
We thank the reviewer for having examined the manuscript very carefully and for the remarks and suggestions of improvements. We have made changes accordingly.
Reviewer 1 comments:
This manuscript describes the structural transitions of triblock copolymer P104 in water covering semidilute to dilute regions as a function of system temperature and polymer concentration. The investigation has been carried out using viscometry, DLS, turbidimetry, microscopy and rheometry. The authors have identified the regions where monomers exist, spherical micelle formation, elongated cylindrical micelles formation, clouding points, and liquid crystalline behavior. The boundaries: liquid-soft gel, soft gel-hard gel, not birefringent/bi-refringent and transparent/opaque, are identified and finally a partial phase diagram temperature-composition of the system in the CP104 range from 0 to 20 wt.%, in which the appearance of the different morphologies was identified: unimers, spherical micelles, cylindrical micelles, worm like micelles. This work seems interesting, but the following minor issues are to be rectified as commented below before publication.
Authors should revise typographical and grammatical errors with better description of unclear sentences for easy understanding.
As recommended by the reviewer, the manuscript was thoroughly reviewed. Typography and grammatical errors were detected and corrected. A better understanding of the paper should be reached.
Page 6, ηo, the o should be as subscript. The equation (6) also needs to be corrected accordingly.
As indicated by the reviewer, equation (6) was corrected, as well as the o as subscript in ηo, page 6.

Reviewer 2 Report
The present manuscript, "Structural Behavior of Amphiphilic Triblock Copolymer P104/Water System" provides a thorough analysis on solution behavior of Pluronic P104. The results are compelling in the context of polymer micellar solution, gelation and biological field. However this study is well reported and well explored from decades so study does not so any novelty. But the techniques used for characterization, depth physical chemistry involved, author's discussion and arguments are adequate, clearly explained, and elegantly written in accordance with the scientific viewpoints. Therefore, I think that this work can be accepted after the following suggestions:
1. In Material section, the order of characterization techniques used is irrespective of next section results and discussion. Please maintain the flow same in both the section. i.e., instrumentation for viscosity measurement comes first as it discussed earlier than others.
2. Symbols represent which concentration is not indicated in Figure S3. Please indicate which line associate with which concentration.
3. Author has specified all the viscosity measured are dynamic. However, author should specify it in Figure S3.
4. What is parameter A and B used in Table 1 from equation 6? Please mention it in manuscript for better understanding to readers.
5. Figure S6 is not cited anywhere in the text.
6. On pg. 17 line “Figure 9d for a sample of 30 wt. %, …..” Here where is Figure 9d comes from? If it is a Figure 9d than provide Figure 9c and d, or if it is typographical error than please correct it.
7. The pattern of references are used are not uniform. Please check it once, i.e., ref. 18.
8. Symbol size and line thickness in Figure 5a need to be increase to see clear indication.
9. Clarity of Figure 8a, b need to improve. insight fig in Fig. 8b is hard to read.
10. Authors can include some recent references of phase behaviour of Pluronics used in drug delivery and several applications, specially in Introduction part and throughout the manuscript. So readers can find novelty in reading and recent uses of pluronics in several applications
Author Response
Answers to reviewer 2:
We thank the reviewer for having examined the manuscript very carefully and for the remarks and suggestions of improvements. We have made changes accordingly and given hopefully convincing replies.
Reviewer 1 comments:
The present manuscript, "Structural Behavior of Amphiphilic Triblock Copolymer P104/Water System" provides a thorough analysis on solution behavior of Pluronic P104. The results are compelling in the context of polymer micellar solution, gelation and biological field. However this study is well reported and well explored from decades so study does not so any novelty. But the techniques used for characterization, depth physical chemistry involved, author's discussion and arguments are adequate, clearly explained, and elegantly written in accordance with the scientific viewpoints.
- In Material section, the order of characterization techniques used is irrespective of next section results and discussion. Please maintain the flow same in both the section. i.e., instrumentation for viscosity measurement comes first as it discussed earlier than others.
The order of the characterization techniques was adapted to maintain the flow in section results and discussion.
- Symbols represent which concentration is not indicated in Figure S3. Please indicate which line associate with which concentration.
The concentrations corresponding to the selected symbols in Figure S3 were indicated as follows:
Figure S3.- Dynamic viscosity of P104 solutions as a function of the temperature at the following CP104: 1 (), 2 (⬠), 3 (), 5 (), 8 (), 10(☆), 15 (✳) and 20 () wt.% Measurements performed at three different angles 30°, 50° and 70°.
- Author has specified all the viscosity measured are dynamic. However, author should specify it in Figure S3.
The viscosity was specified as dynamic viscosity.
- What is parameter A and B used in Table 1 from equation 6? Please mention it in manuscript for better understanding to readers.
Parameters A and B used in Table 1 are now described as follows:
Table 1. Parameters A and B of Equation 1 obtained by fitting dynamic viscosity experimental data. A and B correspond to the empirical constants in Equation 1. B value represents the energy (kJ/mol) necessary for the fluid to start to flow.
- Figure S6 is not cited anywhere in the text.
Figure S6 is cited in the section 3.7 (page 18) of the text as following:
Figures S6 a and b show the optical micrographs of P104/water samples having the concentrations of 50 and 30 wt.%, analyzed at the temperatures of at 80 and 74 °C, respectively. The observed textures are characteristic of hexagonal liquid crystals [75,76].
- On pg. 17 line “Figure 9d for a sample of 30 wt. %, …..” Here where is Figure 9d comes from? If it is a Figure 9d than provide Figure 9c and d, or if it is typographical error than please correct it.
The typographical error was corrected:
Finally, a new behavior of P104 solutions is presented in Figure 10d for a sample of 30 wt.%,
- The pattern of references are used are not uniform. Please check it once, i.e., ref. 18.
All the references were verified, corrected and homogenized.
- Symbol size and line thickness in Figure 5a need to be increase to see clear indication.
Symbol size and line thickness in Figure 5a were increased, as suggested by the reviewer.
- Clarity of Figure 8a, b need to improve. insight fig in Fig. 8b is hard to read.
Figures 8a, 8b, as well as the insert in figure 8b were improved. We believe that now it will be easier to read them.
- Authors can include some recent references of phase behaviour of Pluronics used in drug delivery and several applications, specially in Introduction part and throughout the manuscript. So readers can find novelty in reading and recent uses of pluronics in several applications
We acknowledge the reviewer for this suggestion, recent references about this topic were added in the manuscript.
